# Early succession on slag compared to urban soil: A slower recovery

**Heng-Xing Zou**[1¤], **Alison E. Anastasio**[2]*, **Catherine A. Pfister**[1]

**1** Department of Ecology and Evolution, The University of Chicago, Chicago, Illinois, United States of America, **2** Program on the Global Environment, The University of Chicago, Chicago, Illinois, United States of America

¤ Current address: Department of BioSciences, Rice University, Houston, Texas, United States of America

* aea@uchicago.edu

**Data Availability Statement:** All relevant data are within the papers and its supporting information files. Raw data are also uploaded to Figshare here: https://figshare.com/s/b1f5158a1ea5030d5a92; DOI: 10.6084/m9.figshare.9883589.

## Abstract

Slag, waste from the steel-making process, contains large amounts of calcium, magnesium, iron and other heavy metals. Because of its composition, high pH and low water retention ability, slag is considered inhospitable to plants. Nevertheless, the spontaneously generated plant communities on slag are surprisingly diverse, but the assembly and structure of such communities are poorly studied. Previous studies suggest reduced rates of succession due to low growth rate and slow accumulation of topsoil. To investigate whether slag communities display similar patterns, we used two former industrial sites on the South Side of Chicago, IL, both with high pH (8–9.2) sand content (80%) and calcium concentration (> 9000 ppm). We removed all vegetation from both slag and non-slag plots to test whether recovery differed over one growing season (4 months). To directly assess plant growth, selected focal species were planted on both sites and harvested. We show that recovery from removal differed at slag and non-slag sites: the recruitment process on slag, measured by percent vegetative cover and number of species in plots, was significantly slower at 6–8 weeks of the manipulation and beyond, suggesting a potential stage-dependent effect of slag on plant growth. Certain slag plots recorded less cover than non-slag plots by >30% at maximum difference. Functional trait analysis found that graminoid and early successional species preferentially colonized slag. Overall, slag plots recovered more slowly from disturbance, suggesting a slow succession process that would hinder natural recovery. However, slag also has the potential to serve as plant refugia, hosting flora of analogous habitats native to the area: one of our industrial sites hosts nearly 80% native species with two species of highest Floristic Quality Index (10). Restoration efforts should be informed by the slow process of natural recovery, while post-industrial sites in urban areas serve as potential native plant refugia.

## Introduction

Human activities have drastically modified natural landscapes, creating many uniquely anthropogenic systems. One such example is the urban ecosystem, which encompasses the

**Funding:** Research was supported by internal University of Chicago funds to AEA and a BSCD Summer Fellowship in Ecology and Evolution (https://college.uchicago.edu/academics/biological-sciences-collegiate-division-summer-fellowship) to HZ. The funders had no role in study design, data collection and analysis, decision to publish, or preparation of the manuscript.

**Competing interests:** The authors have declared that no competing interests exist.

biological activities and ecological processes that are contained in and emerge from a built environment. With high disturbance, largely patchy habitat, and highly variable soil composition, urban ecosystems generally have unique community dynamics [1]. Among many urban habitats, sites where industrial dumping has occurred have gained attention from both ecologists and the public because they raise unique environmental and public health concerns [2, 3]. Slag, a byproduct of steel production that contains a large amount of heavy metals and alkaline earth metal compounds [4], is used as fill or simply dumped in steel-making areas. This industrial waste is estimated to comprise over 52 km$^2$ of land surface in northeastern Illinois and northwestern Indiana in the greater Calumet region [4]. Slag sites, often originally water bodies, are filled or covered by a layer of slag material varying in size from small granules to big chunks to large contiguous surfaces; depths of slag fill reach up to 18m [5]. The chemical composition, as well as thin topsoil, low water retention, and low organic matter, makes slag sites a generally inhospitable place for plant life [4, 6]. Although variable in specific contents, steel slag is generally comprised of calcium and magnesium oxide and silicate, compounds of iron, manganese and other heavy metals [4]. Some slag contains organic pollutants such as polycyclic aromatic hydrocarbons [6, 7]. Because of the high calcium and magnesium content, the pH of slag is generally basic. Depending on the specific composition of slag, some contents may leach into nearby water bodies, causing significant pollution [2]; even more strikingly, Big Marsh Park in the Calumet region contains a pond with pH higher than 12 [6].

Due to its history as a center for steel production, the Calumet region, defined by the Calumet River watershed in Illinois and Indiana, has experienced the extensive dumping of slag. Typically, slag and other waste was dumped on land adjacent to industrial plants, into pits, lakes or wetlands, or used as fill [5]. Remediation of industrial dumps of the Calumet region is desired as sources of contamination has always been a major public health concern [2, 4, 6]; therefore, multiple slag sites in the region have undergone different degrees of reclamation. Some, including parts of the Big Marsh Park, have been sites for active ecological remediation and have been transformed into vibrant urban parks [8], while others such as U.S. Steel South Works have remained relatively untouched for decades [9]. Common methods to prepare sites in the region for ecosystem construction include phytoremediation or capping with organic material [7, 10]. Nevertheless, it has been observed that because of their mobility, contaminants in slag affect plant growth even with addition of compost, and experiments with different plants used for phytoremediation resulted in low survival [6, 7]. Furthermore, in some cases, other debris such as construction and demolition waste has also been dumped at these industrial sites, complicating the land use history and making reclamation more difficult.

Slag sites, however, can harbor diverse vegetation, and plant communities at some sites are comparable to those in urban disrupted land or vacant lots, with many non-native and weedy grasses, forbs and shrubs. Native forbs and grasses are also common. In general, few trees successfully establish on slag, but those that do include cottonwood (*Populus deltoides*), mulberry (*Morus alba*), and staghorn sumac (*Rhus typhina*). Typical wetland weedy species, such as several common sedges, common reed (*Phragmites australis*), and cattails (*Typha spp.*) are found near depressions on slag. A number of native species of high conservation value have also been observed, including whorled milkweed (*Asclepias verticillata*), elliptic spikerush (*Eleocharis elliptica*) and nodding lady's tresses (*Spiranthes cernua*). Overall, given the harsh environment of slag as a substrate, the spontaneous plant communities at some sites are surprisingly diverse.

Naturally, such spontaneous vegetative communities raise the question of how they have assembled. According to classical successional theory, soil formation is enhanced by primary successional species that can facilitate succession by other species [11]. Generally, drought- and heat-resistant, primary successional species grow fast, producing large amounts of litter

that integrate into the soil, increasing organic material content [11–13]. Therefore, the ability of the primary successional community to provide organic material for soil is important for both the survival of later successional species and the recovery of an ecosystem from disturbance (facilitation model [14]). Without soil formation, many natural habitats display extremely slow or arrested primary succession. For instance, alvar habitats, comprised of thin topsoil on limestone bedrock, have high pH, shallow topsoil, and low nutrient availability similar to slag; due to the low growth rate of early successional plants, the community is "arrested" at a primary successional stage and does not succeed to later stages [15, 16]. Similarly, reclamation efforts of forests on old mine sites are incomplete due to the failure of late successional and specialist species to colonize [17, 18].

A slag site can be considered a habitat after disturbance, with most plants killed and soil covered by dumping. As disturbance ends (i.e. no further dumping) and erosion breaks the contiguous slag surface, primary succession takes place and the present community assembles. Previous studies have estimated successional trajectories of other industrial sites, predicting recovery to the original climax community in 75–150 years [19, 20]. However, it is possible that plant communities on slag, though they have been relatively undisturbed for long periods of time, may have arrested primary succession. Because slag sites are usually characterized by a thin layer of topsoil with high pH, they might be comparable to alvars in displaying "arrested" soil and plant community development. The environmental stress on slag may lower growth rates of early successional plants, resulting in less accumulated plant litter and decreased soil formation [21]. This hypothesis is supported by the fact that some sites, including Big Marsh and Van Vlissingen Parks, have been undergoing succession from 1977 and 1927, respectively [5], but could not support survival of later-successional species, such as most trees and shrubs, except with an experimental supplement of compost [6]. Nevertheless, more evidence is needed to determine whether such slag plant communities are arrested in early successional stages by the lack of nutrients.

We tested the following hypotheses regarding the different succession process on slag: that 1) slag sites differed from non-slag sites by soil composition and community structure, 2) the recovery rate from disturbance, measured in percent cover, growth rate and number of recolonized species, differed for slag and non-slag sites, and 3) plants on slag differed in their functional traits that reflect adaptation to early successional habitats. Over the course of one growing season (June to October 2018), a combination of plant surveys, soil testing and controlled experiments on two types of sites, **Slag** and non-slag or urban soils (hereafter **Reference**), were used to test the three hypotheses. Overall, we expected to see a slower recovery of plants and a non-random assembly of early successional communities on slag.

## Methods

### Site selection and characterization

The study was conducted at two locales: Big Marsh Park (**BM**) and Van Vlissingen Park (**VV**), both Chicago Park District properties located on Chicago's Southeast Side (**Fig 1**; permit #1766). Neither site contains any remnant natural habitat but is part of a larger, highly modified wetland complex. The locales were chosen so that they both contained Slag (**S**) and non-slag Reference (**R**) sites, where R sites had urban soil deposited on top of fill of mixed origins [5], and were in close proximity to each other (within 1.9 km), to minimize variance in local climate. Slag at Big Marsh (**BM-S**) was deposited between 1965–1977 while slag at Van Vlissingen (**VV-S**) was deposited between 1902–1927 [5]. The reference site at BM (**BM-R**) has been seeded and managed for invasive species by Chicago Park District (See **S1 File** for details), and **VV-R** is surrounded by sparsely spaced cottonwood (*Populus deltoides*). All sites

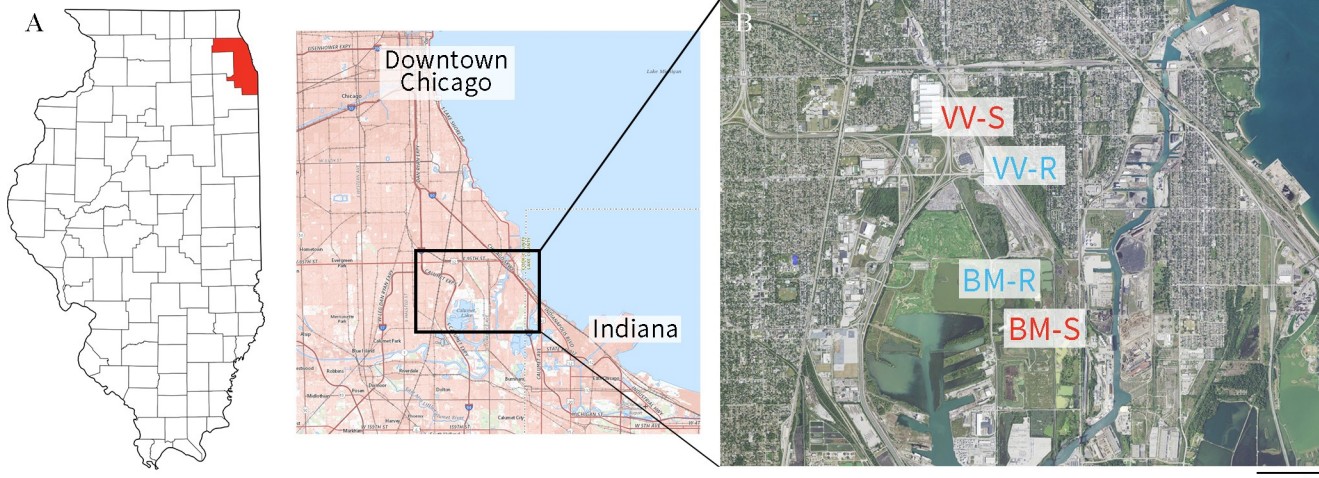

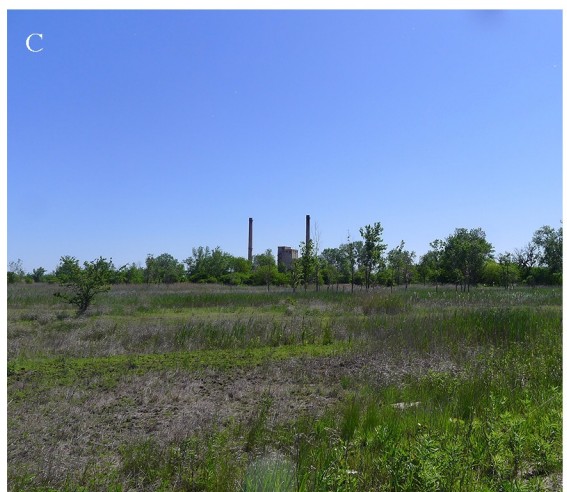

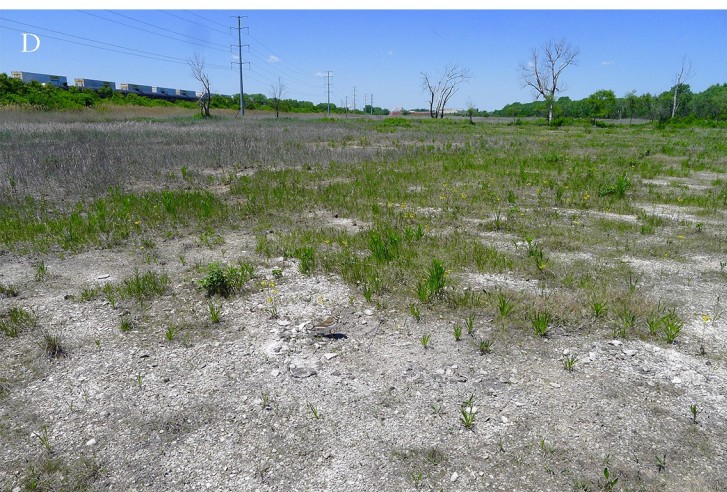

**Fig 1. Maps and photos showing experimental site.** (A) Map of Illinois and a regional map showing location of sites. (B) Zoomed-in satellite image showing Big Marsh (BM) and Van Vlissingen (VV) Parks. (C) Photo of slag site at BM. (D) Photo of slag site at VV. Maps reprinted from [22] under a CC BY license. Site photographs taken by the authors. Site coordinates: BM-S (41.686, -87.567), BM-R (41.694, -87.575), VV-S (41.711, -87.576), VV-R (41.709, -87.573).

are in full sun, except for a few plots at VV-R which encounter partial shade for no more than 2 hours per day. Both slag sites contain slight depressions which allow standing water to accumulate after heavy rainfall. The experimental site at VV-S is surrounded by *Phragmites*-dominated shallow slag-bottomed wetlands.

The study spanned 4 months, from Jun 5, 2018 (planting focal species) to Oct 6, 2018 (final harvest). Weather data of study sites were drawn from the National Weather Service [23]. Highest temperature ranged from 17.8 (Jun 22) to 36.1°C (Aug 4), and low temperature ranged from 8.9 (Jun 6) to 24.4°C (Jul 1). Highest precipitation was 6.00 cm (Aug 7). Typically, weather at the study sites was highly variable temporally and geographically; during the study period, the weather was characterized by consecutive hot and sunny days, continuous rainfall over prolonged periods and occasional local thunderstorms.

Plant surveys were conducted on Jul 9, 2018 at VV-S and Jul 31, 2018 at BM-S using the censusing procedure from the Northwest Indiana Restoration Monitoring Inventory (NIRMI). At each site, a 50 m × 20 m survey plot was set up, and a comprehensive list of

species and cover data was recorded according to NIRMI procedure [24]. At BM-S, parts of the experimental plots were included in the survey. Species lists produced at BM-R and VV-R were in close proximity of experimental plots (≤ 10m), providing a regional species list for identification of species potentially germinating in experimental plots. Floristic Quality Assessment (FQA, [25]) was also performed on species lists using the Universal FQA Calculator [26] to evaluate conservation values at each site. FQA is used to estimate habitat quality based on the coefficient of conservation for each species present; the coefficient is determined by whether the species is disturbance-adapted (low score) or associated with undisturbed natural area (high score; [25]).

Samples of soil at each site were drawn from 10 different spots < 0.5 m from experimental plots, with an approximate depth of 10 cm. Samples were also taken from the commercial topsoil used in germination plots (see Experimental Setup: *Experiment 1*: *Germination*). Two distinctive types of soil were observed at VV-S, and were therefore sampled separately and designated VV-S-1 (block A-D) and VV-S-2 (block E). Samples from each site were combined and ~ 400 mL soil was drawn from each mixture for laboratory analysis. A total of 6 samples (one per each site for BM-S, BM-R, VV-R; two for VV-S; one from commercial topsoil) were analyzed by A&L Great Lakes Laboratories (Fort Wayne, IN) using the environmental variables in **Table 1**.

## Experimental setup

The basic unit of replication was the block, which consisted of five types of experimental plots to evaluate two general aspects of plant performance: three examined growth of established focal species and two tested germination success (**Fig 2A**). Each site hosted five blocks, named A-E. The order of plots within block was randomized. Two blocks on each Reference site contained only three focal species plots due to spatial constraints. Overall, each Slag and Reference site hosted 25 plots and 21 plots, respectively.

**Experiment 1: Germination.** To evaluate which species are involved in recolonization after disturbance, all above-ground vegetation was removed from experimental plots (1.0 × 1.0 m, with additional buffer zone of 0.2 m on each edge), followed by either tilling and removal of 5–10 mm of extant soil and plant material (hereafter **removal**) or removal plus covering by 5-mm thick layer of commercial topsoil (New Plant Life All Purpose Topsoil, Markman Peat Corp., Le Claire, IA; hereafter **topsoil**). Removal plots simulated the recruitment of plants into newly disturbed habitat with no current residents; topsoil plots characterized the "background dispersal rate" because the effect of growth from seed bank or below-ground vegetative structures remaining in the soil was reduced.

To measure the establishment and growth rate of plants in germination plots, monitoring photos were taken weekly during the course of 16 weeks. An aluminum quadrat (0.25 × 0.25 m) was placed at one corner of the 1.0 × 1.0 m plot area, and a photo was taken for each corner. Each photo was then imported to ImageJ [27]. The percentage of green vegetation within quadrat (hereafter **cover**) was extracted; cover of a plot was then determined by the arithmetic average of cover values from all four photos.

**Table 1. Variables measured by soil test, in four major categories.**

| Category | Variables Measured |
|---|---|
| Nutrient Level | Organic matter (percentage), N, P, K |
| Chemical Properties | Mg, Ca, pH, Cation Exchange Capacity (CEC) |
| Heavy Metal Content | Zn, Mn, Fe, Cu, As, Cr, Pb |
| Physical Properties | Sand, silt, clay (all percentages) |

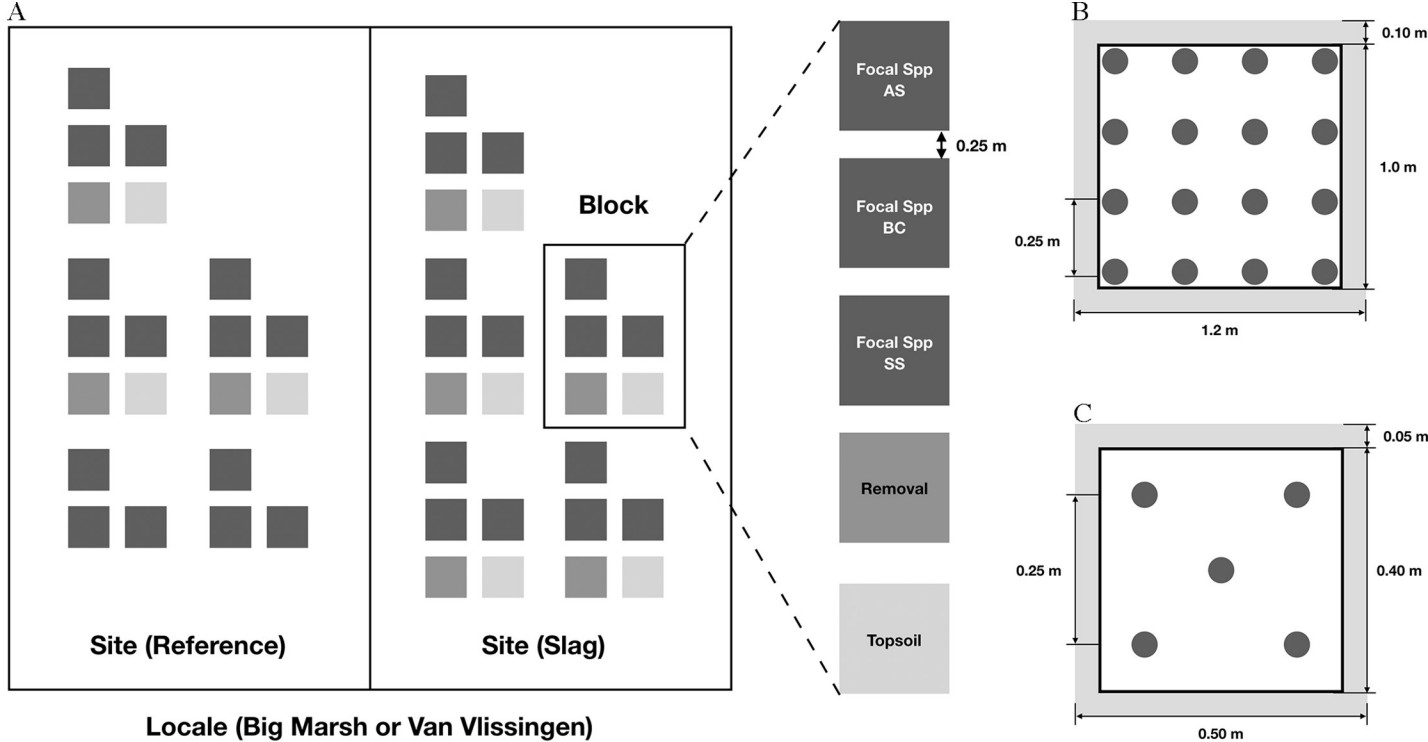

**Fig 2. Experimental setup and plot maps.** (A) General map showing number of plots, and relationships between locale, site and block, with configuration within each block; each plot is randomly ordered and separated by at least 0.25 m. (B) Configuration of 16 individuals within each BC and SS focal species plot. (C) Configuration of 5 individuals within each AS focal species plot.

To calculate species diversity within germination plots, species presence/absence data for experimental plots was recorded from weekly monitoring photos selected from three dates, Aug 22, Sep 5 and Sep 26 (day 77, 91 and 112). A species was considered present if observed in any of the three records. To discover possible time trends in diversity patterns, the number of species present in each plot over the whole experimental period was also counted and recorded from monitoring photos.

**Experiment 2: Focal species.** To directly measure the growth rate of plants on Slag and Reference sites, three native focal species with high conservation value were manually planted in experimental plots. For sideoats grama (*Bouteloua curtipendula*, **BC**) and showy goldenrod (*Solidago speciosa*, **SS**), each $1.0 \times 1.0$ m plot (with buffer zone of 0.2 m on each edge) hosted 16 plants in $4 \times 4$ pattern (**Fig 2B**); due to a sourcing constraint, only five common milkweed (*Asclepias syriaca*, **AS**) were planted in a $0.5 \times 0.5$ m plot (with buffer zone of 0.1 m on each edge; **Fig 2C**). Seedings less than a month old and were sourced from Cardno Native Plant Nursery (Walkerton, IN). All focal plants were watered daily during the first two weeks and at least once a week afterwards. No other manipulations (intensive weeding, application of fertilizer or pesticide) or physical protection were done.

## Biomass harvest

The aboveground biomass was quantified twice for focal species plots and once at the end of season for germination plots. For focal species plots, an initial harvest was done at week 2 (Jun 20, 2018) where two plants (one for AS plots) were harvested for each focal species plot, and a final harvest was done at week 17 (Oct 6, 2018), when all plants in focal species plots were

harvested except those harvested art week 2 and re-sprouted. Therefore, a maximum of 14 (4 for AS plots) plants were harvested from each plot, with the exact number depending on the survivorship. For germination plots, the central 0.25 × 0.25 m portion was harvested so that an equal portion of each corner that contributed to the cover measurement was included. After harvest, all biomass was left at room condition for no more than 12 hours before drying at 80˚C for at least 96 hours. The biomass in germination plots was measured as total biomass of all species in the plot; each focal species plant was measured individually.

## Data analysis

All data analyses and plotting were done in R version 3.5.1 [28]. See **Supplementary Methods** in **S1 File** for a full list of packages used.

**Soil, cover and biomass.** Soil data were scaled, centered, and analyzed by principal component analysis (PCA). The resulting sets of vectors characterizing each soil sample were then extracted as independent variables in a linear regression of cover and biomass.

We tested whether site, locale or treatment of germination plots affected cover and biomass by one-way ANOVA. The relationship between cover and species number at each timepoint in removal plots was investigated by linear regression. Because the experiment involved a block design, linear mixed effects (LME) models were also fitted for block effect as a source of random error. Biomass estimates were divided into three groups: initial harvest, final harvest of focal species, and germination plots. All groups were analyzed using Welch's t-test and Wilcoxon's test when applicable, with biomass on Slag lower than on Reference as the alternative hypothesis. Data from final harvest of germination plots were further analyzed with two-way ANOVA of site and treatment effects.

We tested whether plant cover and biomass were related to soil properties using linear regression, with soil measurements extracted from PCA as independent variables. To compress the time series data of cover, the maximum was taken for each plot. Missing samples due to mortality were designated 0 in biomass analyses. Linear regression analyses used a Bonferroni correction; with 16 soil measurements, the adjusted $p$ value was 0.05/16 = 0.003125.

**Species diversity and community composition.** Species presence-absence data from germination plots and on both Slag sites was compiled from three days of monitoring photos and the plant survey, respectively (see *Experiment 1*: *Germination* and **S1 File**). For both datasets, species richness ($\alpha$ diversity), plot dissimilarity (Whittaker's $\beta$ diversity; [29]) and native status were analyzed. Non-metric multi-dimensional scaling (NMDS) analysis was performed on presence-absence data in germination plots to further evaluate the dissimilarity among sites and locales using the R package *vegan* [30]. Species number from each plot over the experimental period was also compiled from monitoring photos and subsequently tested for any site effects using ANOVA.

We also tested whether functional traits differed in communities that colonized removal plots on Slag and Reference, selecting traits that are important to plant production and reproductive strategies ([13, 31, 32]; **Table 2**). Functional traits and native status for species present in germination plots were obtained from Hilty [33], the TRY database [34] and Grime et al. [35]. Canonical (constrained) correspondence analysis (CCA; [36]) was used to detect the potential correspondence between environmental variables and species distribution in germination plots. Model selection was based on maximum likelihood method using the R function *ordistep* in the package *vegan* [30] to obtain environmental variables that best explain species composition. To evaluate the association between functional traits and environmental variables, an RLQ model [37] was constructed with the package *ade4* ([38]; see **S1 File** for details).

**Table 2. Plant functional traits used for analysis.**

| Trait Category | Trait Name |
|---|---|
| **Lifestyle** | Species type (herb, grass, vine, shrub, tree) |
| | Life history (annual, biennial, perennial) |
| | Growth form (grass, tall forb, short forb, shrub, tree) |
| | Functional group (graminoid, forb, legume, woody) |
| **Physiology** | Growth habit (erect, decumbent, procumbent, sprawling, vine) |
| | Shoot structure (leafy, semirosette, rosette) |
| | Canopy length (quantitative, in cm) |
| **Regeneration** | Lateral spread (<0.01 m, 0.01–0.25 m, >0.25 m) |
| | Regenerative strategy (widespread seed, vegetative spread, seasonal by seed) |
| | Seed number (quantitative) |
| | Seed dry mass (quantitative, in mg) |
| | Seedbank longevity (short: under 1 yr; medium: 1 to 5 yrs; long: >5 yrs) |
| | Phenology (early: before June; summer: June to July; late summer: after July) |
| **Primary Production** | Leaf dry matter content (LDMC; quantitative, in mg/mg) |
| | Specific leaf area (SLA; quantitative, in mm$^2$/mg) |
| **Native status** | Native or Nonnative |

## Results

### Plant survey of slag sites

Slag sites at BM and VV hosted 66 and 44 species, respectively. Most species were herbaceous and relatively short in stature, forming a sparse cover. Dominant species included rough false pennyroyal (*Hedeoma hispida*), *Dichanthelium spp.* and rosette-forming forbs including flea-banes (*Erigeron spp.*), goldenrods (*Solidago spp.*) and plantains (*Plantago spp.*). Invasive European buckthorn (*Rhamnus cathartica*) and glossy buckthorn (*R. frangula*) are examples of shrubs on slag. Although seedlings of several trees were observed sporadically on slag, the only established trees are cottonwood (*Populus deltoides*), mulberry (*Morus alba*) and sumac (*Rhus spp.*; not in surveyed area); trees display visibly stunted stature compared to those growing on non-slag soil. Many invasive wetland species, such as common reed (*Phragmites australis*), cattail (*Typha × glauca*), and purple loosestrife (*Lythrum salicaria*) have either established monoculture or contribute largely to the plant community in wet depressions on slag.

FQA showed that VV-S is a higher quality habitat by hosting more native and high-conservation value species than BM-S. At BM-S, percentage of native species was 55.6%; the adjusted Floristic Quality Index (FQI) for the community was 19.4, with *Solidago rugosa* having the highest FQI of 6. At VV-S, percentage of native species was 79.4%; the adjusted FQI for the community was 40.1, with *Carex crawei* and *Eleocharis elliptica* both having the highest FQI of 10. See **S2 File** for the full species list and associated FQI.

We did not conduct surveys on both Reference sites because our focus was on slag communities. Reference sites have been seeded and/or undergone intense management by Chicago Park District. We have obtained lists of planted species on BM-R and a basic plant survey of VV-R (together in **S10 File**) conducted by Chicago Park District.

### Soil characteristics

**Table 3** shows the main results of soil analysis. pH and Ca content of both Slag sites are considerably higher than those of both Reference sites (8–9.2 comparing to 7.2–7.9). Sand contents are also higher on Slag sites (72–80% compared to 50%), indicating lower water retention

**Table 3. Soil test results of samples from slag and reference sites.**

| Measurement | BM-R | VV-R | BM-S | VV-S-1 | VV-S-2 |
|---|---|---|---|---|---|
| **pH** | 7.9 | 7.2 | 9.2 | 8.2 | 8 |
| **Ca** | 2150 | 3850 | 10700 | 10150 | 9350 |
| **Mg** | 505 | 430 | 435 | 160 | 165 |
| **Fe** | 68 | 42 | 51 | 13 | 53 |
| **Mn** | 36 | 59 | 225 | 109 | 143 |
| **Zn** | 18.9 | 41.4 | 70.6 | 10.5 | 26.5 |
| **Cr** | 20.5 | 35.3 | 227 | 15.6 | 19 |
| **Cu** | 6.3 | 9.9 | 3.6 | 1.9 | 2.7 |
| **Pb** | 79.4 | 353 | 147 | 38 | 181 |
| **As** | 9.8 | 7.99 | 6.11 | 3 | 3.63 |
| **P (Bray, Total)** | 18 | 20 | 22 | 3 | 3 |
| **N (Total)** | 8 | 21 | 33 | 5 | 8 |
| **K** | 106 | 175 | 108 | 258 | 281 |
| **Sand (%)** | 50 | 50 | 80 | 72 | 78 |
| **Clay (%)** | 24 | 22 | 6 | 8 | 6 |
| **Silt (%)** | 26 | 28 | 14 | 20 | 16 |
| **Organic Matter (%)** | 6.1 | 6.6 | 5.8 | 2.3 | 3.4 |

Two samples were obtained from VV-S, denoted as VV-S-1 and VV-S-2. Units for elemental concentration is parts per million (ppm). See **Table A** in **S1 File** for full soil test results.

rate. BM-S had higher N and P content than all other sites and an organic matter content comparable to other Reference sites, in contrast to the expectation that slag sites are lower in nutrients. Heavy metals showed no consistent patterns between Slag and Reference sites, also counter to our expectation.

PCA of soil measurements showed significant differences between Slag, Reference and commercial topsoil (CTRL; **Fig A** in **S1 File**). The first two principle components explained 42.99% and 29.57% of the total variance. Specifically, Slag samples were characterized by high Ca, Mn, K, Zn, Cr, sand content and higher CEC and pH (**Table 4**).

## Slag effects on recovery

**Cover.** Overall, Slag plots showed lower plant cover than Reference plots at both locales. ANOVA on cover showed a significant site effect between Slag and Reference with both treatments at both locales (**Table 5**), but the effect of site was not distinctive until later in the experiment (day 49 to 63), as shown by time series plots of cover between sites of the same locale and treatment (a "time threshold"; **Fig 3A–3D**; **Table C** in **S1 File**). Removal plots on both slag sites showed slower increase of cover with increasing species number, indicated by smaller slope of regression lines (**Fig 4**; **Table E** in **S1 File**). It is worth noting that there was a significant locale effect on cover across sites and treatments, except between removal plots on

**Table 4. Characterization of soil samples by measurements from PCA.**

| Category | Variables |
|---|---|
| **I. Slag** | Ca, Mn, K, Zn, Cr, Sand, CEC, pH |
| **II. Reference** | Cu, Pb, Mg, Clay, Silt |
| **III. Topsoil (CTRL)** | Organic matter, P, N, As, Fe |

**Table 5. ANOVA results of site (between slag and non-slag sites) and locale effects (between BM and VV) on cover in germination plots.**

| | Site Effect, BM | | Site Effect, VV | | Locale Effect, Slag | | Locale Effect, Reference | |
|---|---|---|---|---|---|---|---|---|
| | Removal | Topsoil | Removal | Topsoil | Removal | Topsoil | Removal | Topsoil |
| *F* | 14.45*** | 42.64*** | 61.99*** | 57.37*** | 86.05*** | 33.19*** | 2.556 | 27.16*** |

All analyses with 1 degree of freedom. Significance levels: $p < 0.05$ (*), $0.01 < p < 0.05$ (**), $p < 0.001$ (***).

Reference sites (**Table 5**), suggesting that the two locales cannot be treated as replicates. Furthermore, using linear mixed-effect (LME) models, both removal and topsoil plots at BM showed a significant block effect ($\sigma_{\text{block effect}}/\sigma_{\text{overall}}$ = 2.422/7.308 and 2.552/8.163, respectively), indicating a high heterogeneity within these sites. See **S5 File** for the raw cover data.

**Species number.** Removal plots on slag hosted fewer species at both locales (ANOVA on site effect, $F$ = 26.18 and 47.82 for BM and VV, respectively; both $p < 0.01$), although the difference between Slag and Reference did not diverge until day 63 at BM, a pattern similar to the "time threshold" reported for the site effect of cover (**Fig 3E** and **3G**, **Table C** in **S1 File**; see **S6 File** for the raw cover data.).

**Biomass.** Harvested aboveground biomass on Slag sites was lower than Reference at both BM and VV for both germination and focal species plots. The difference was significant for germination plots using Welch's t-test on log-transformed data (**Fig 5D–5E**; **Table F** in **S1 File**). Two-way ANOVA on site (Slag or Reference) and treatment (removal or topsoil) showed a consistent effect of slag ($F$ = 21.335 and 33.53, respectively; both $p < 0.001$) but not treatment (between removal and topsoil; see **Table G** in **S1 File**).

During initial harvest, focal species plots showed little effect of site or locale: while the biomass of BC on Slag was higher than on Reference (both Welch's and Wilcoxon's tests, $p < 0.05$); the significance was not robust using log-transformed biomass data (**Table F** in **S1 File**). At the end of the growing season, the biomass on Slag sites was consistently lower than on Reference soils for BC and SS using both Welch's and Wilcoxon's tests ($p < 0.01$; **Fig 5B** and **5C**). Notably, the high mortality (43/70, or 61.4%) of SS on BM Slag resulted in a median biomass of 0. The strong effect of slag was still detectable using only surviving SS biomass. The high mortality of AS across all sites and locales (54/80, or 67.5%) led to little difference in final biomass and therefore was not informative (**Fig 5A**). No mortality was observed for BC at any site. BC at both BM and VV and SS at VV showed a significant block effect ($\sigma_{\text{block effect}}/\sigma_{\text{overall}}$ = 0.7642/1.929, 1.382/3.538 and 0.38/1.607, respectively).

**Correlation with soil measurements.** Slag soil variables mostly correlated negatively to plant growth, measured in both cover and biomass when cover and biomass were fitted to each soil variable with linear regression (LR) models using both untransformed and log-transformed soil measurements. The transformation of soil variables yielded no qualitative differences on LR results. With only a few exceptions, most variables in category I (Slag; see **Table 4**) negatively correlated with plant growth, measured in either cover or biomass, of both experiments; most variables in category II (Reference) and III (Topsoil) positively correlated with plant growth (**Table J** in **S1 File**).

## Structure of recolonized communities

In the original experimental design, topsoil plots were set up to minimize germination of the local seed bank and to characterize the background dispersal rate. However, the commercial topsoil was not sterile, biasing species diversity and community composition estimates. Analyses on community structure (see below) excluded data from topsoil plots (see **Supplementary Methods** in **S1 File**).

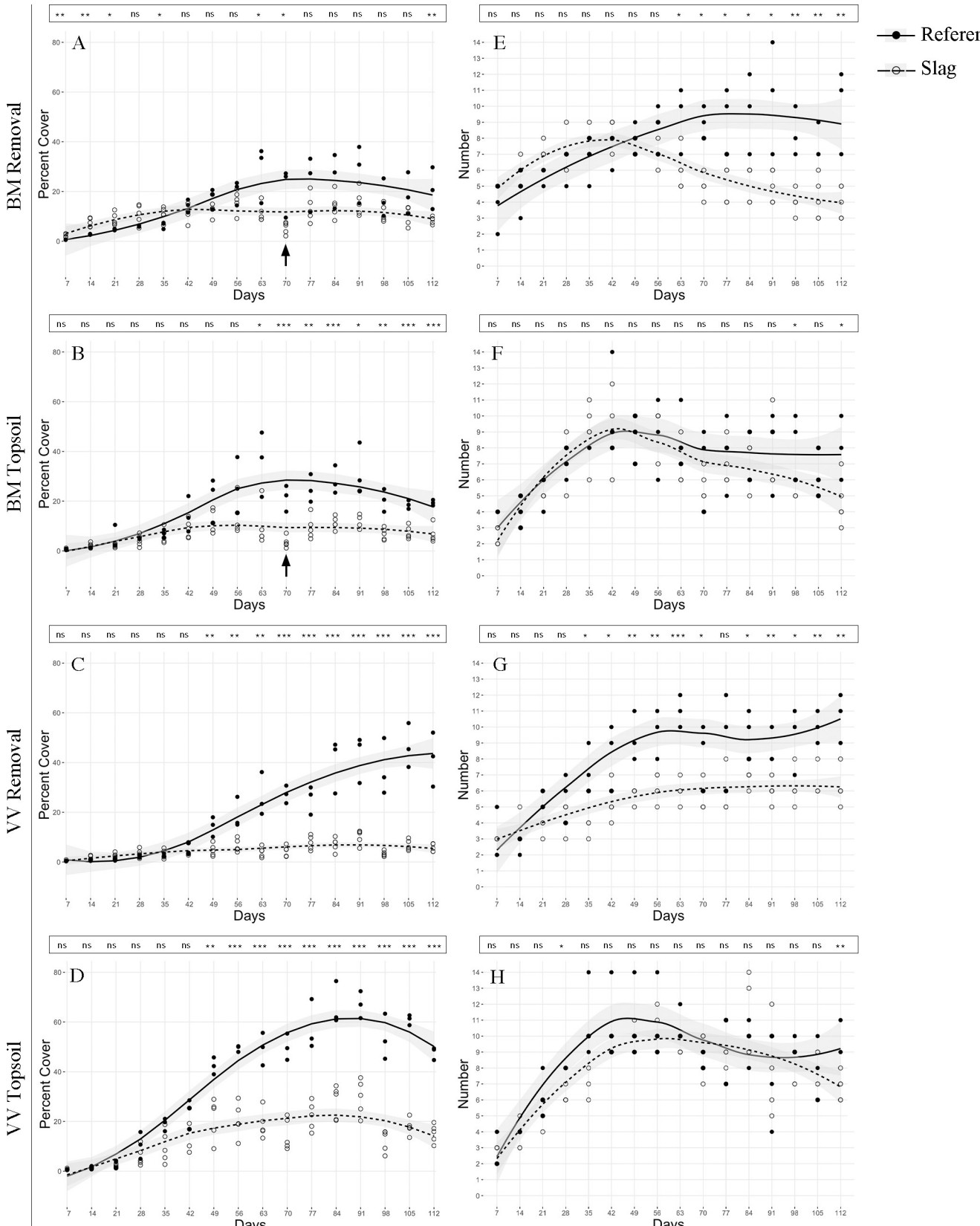

**Fig 3. Effect of slag on percent cover and species number in germination plots over time, in days.** (A)-(D) Mean percent cover. (E)-(H) Species number. Percent cover and the number of species in germination plots between Slag (dashed line) and Reference (solid line) sites at each locale are plotted. $x$ axes show time from start of experiment in days. Shaded areas denote one standard deviation. A dip in cover at Big Marsh can be seen at day 70, shown by arrows. Notation above each graph shows the site-effect ANOVA result using data from that specific day. Significance levels: $p > 0.05$ (ns), $p < 0.05$ (*), $0.01 < p < 0.05$ (**), $p < 0.001$ (***). See **Tables C-D** in **S1 File** for complete statistics.

**$\beta$ Diversity.** $\beta$ diversity (as Whittaker's $\beta$) showed that removal plots on the same site had high similarity (**Fig B** in **S1 File**), in contrast to Slag or Reference plots across locales. Removal plots differed in species composition across sites (**Fig 6A**). Permutational Multivariate ANOVA (PERMANOVA) results suggested a significant difference between four sites ($R^2 = 0.726$, $p = 0.001$) but not among blocks within each site (see **Table H** in **S1 File** for full statistics). **Fig D** in **S1 File** shows the distribution of species and sites in NMDS space. Most species resided in clusters, indicating their constrained distribution at some sites; cottonwood (*Populus deltoides*, *PODE*) and false pennyroyal (*Trichostema brachiatum*, *TRBR*) were present in plots across multiple sites and therefore were distant to all existing site clusters.

**Functional traits.** Results from CCA showed significant clustering for plant species (**Fig 6B**). Similar to the results from NMDS, each cluster of plots had a distinctive collection of species. Model selection using all environmental variables identified soil K, Mn and N as the three most significant variables determining species distribution. Limiting environmental variables to category I (Slag) and category II + III (Reference; see **Table 4**) yielded different results: for Slag, a collection of soil Ca, pH and Mn explained most of the variation in species distribution; for Reference, the explanatory environmental variables changed to organic matter, N and Mg.

Fourth-corner analysis showed some signal of environmental filtering on functional traits (**Fig 6C**). Graminoid species (*Funct.G*) and species with widespread seeds (*Regen.W*) associated most strongly with organic matter, P, Mg, Fe, As (negative) and K (positive). Late-summer flowering species (after July; *Pheno.L*) were positively associated with soil N, Zn and Cr; the reverse was true for summer-flowering species (before July; *Pheno.S*). Specific leaf area (SLA) was negatively associated with soil pH and Cr and positively associated with K. Although some individual associations were significant, the overall association was not significant as determined by the $S_{RLQ}$ statistic ($p > 0.50$). Seed bank longevity and lateral spread

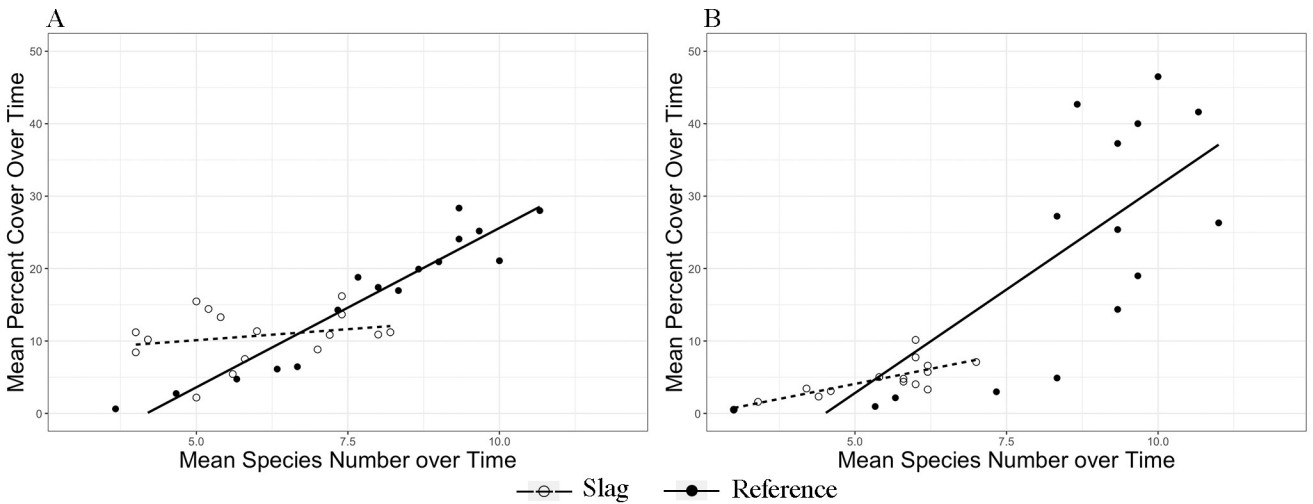

**Fig 4. Linear regression of mean percent cover versus mean species number in germination plots.** (A) BM. (B) VV. Slopes of regression lines are different for plots on Slag (dashed line) and Reference (solid line) sites. See **Table E** in **S1 File** for complete statistics.

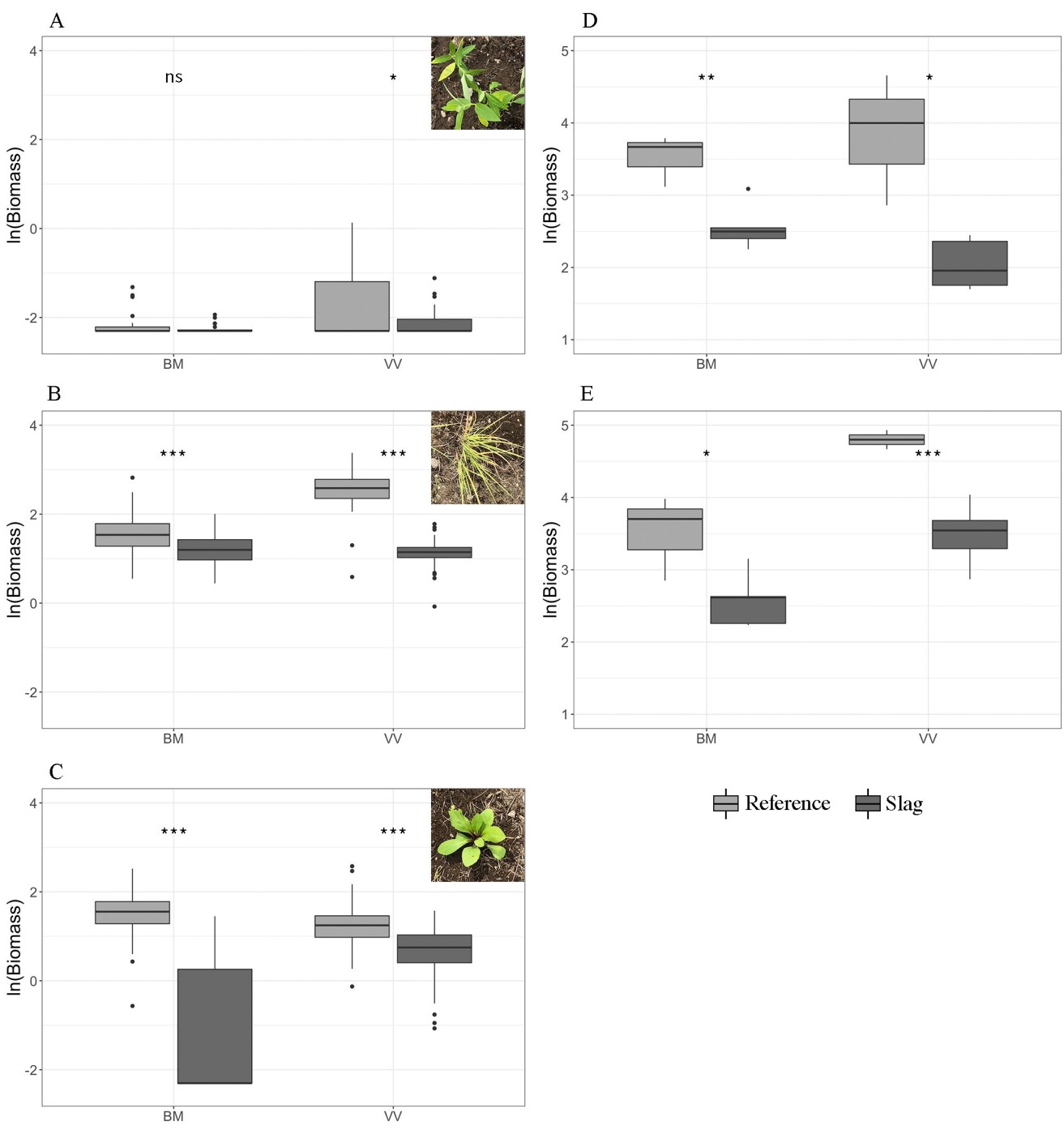

**Fig 5. Comparison between locales and sites of final harvested biomass, in grams, from all types of experimental plots.** (A) Focal species *Asclepias syriaca*. (B) Focal species *Bouteloua curtipendula*. (C) Focal species *Solidago speciosa*. (D) Removal plots. (E) Topsoil plots. All *y* axes are ln(biomass + 0.1) in *g*, *x* axes are locales. Significance levels: $p > 0.05$ (ns), $p < 0.05$ (*), $0.01 < p < 0.05$ (**), $p < 0.001$ (***). See **Table F** in **S1 File** for complete statistics. All photos were taken by the authors.

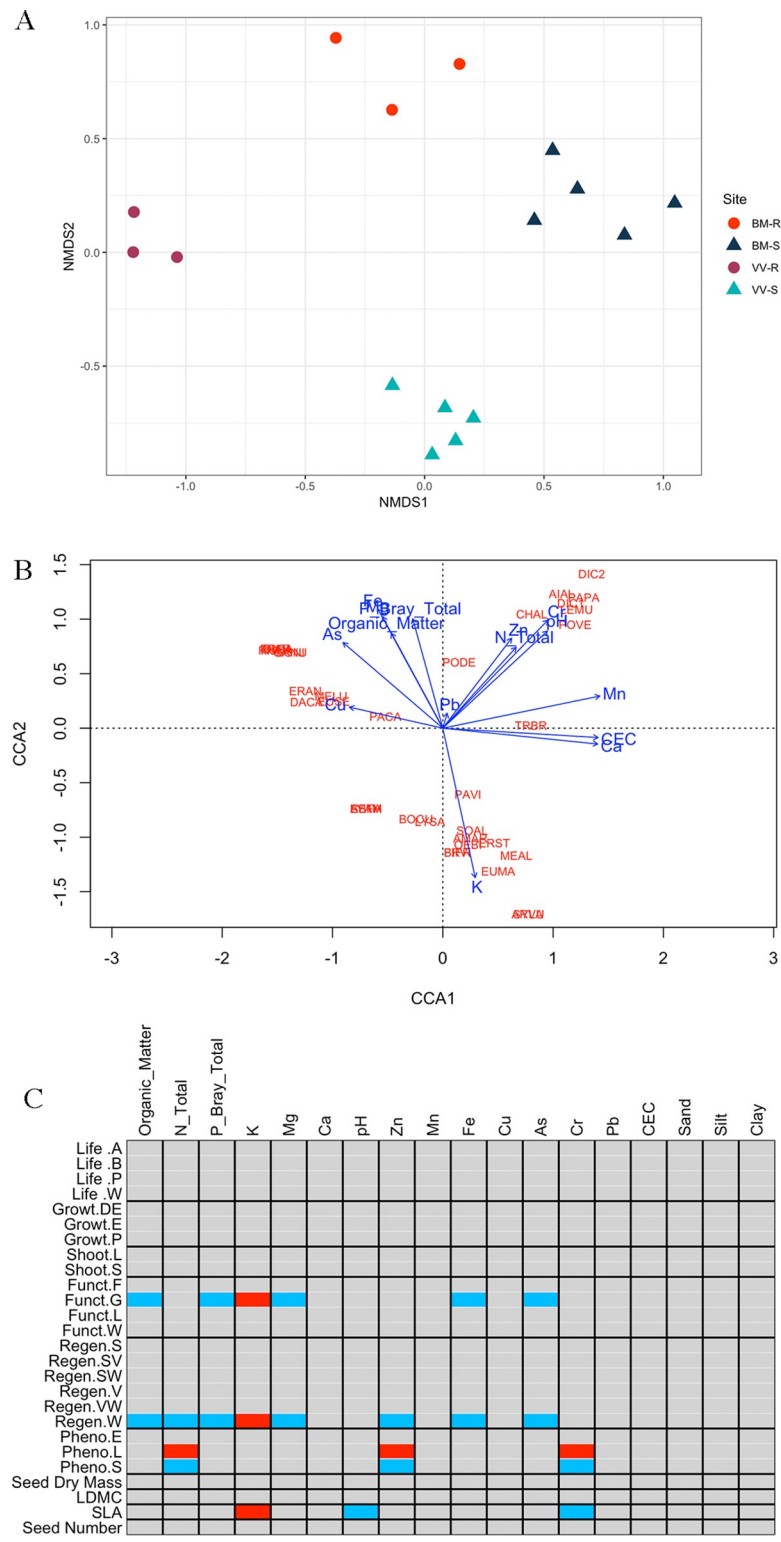

**Fig 6. Community structure of removal plots.** (A) NMDS results of site difference based on species presence-absence. Slag plots are in triangles, Reference plots in circles. See **Table H** in **S1 File** for associated statistical analyses, and **Fig D** in **S1 File** for inclusion of species in the NMDS space. (B) CCA result showing association between species and environmental variables. (C) Fourth-corner analysis of RLQ model showing association between functional traits and environmental variables; significant associations are colored, with positive as red and negative as blue. See **S3 File** for a complete list of species abbreviations; see **S4 File** for complete functional trait data with abbreviations.

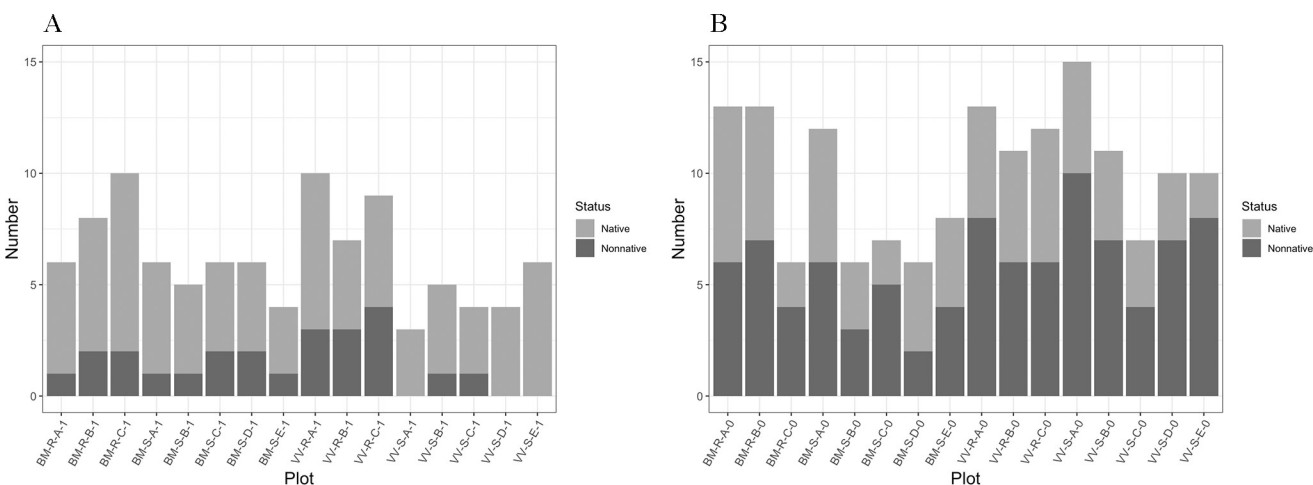

**Fig 7. Native status of species in germination plots.** (A) Removal plots. (B) Topsoil plots.

were excluded from this analysis because of not enough data. See **S4 File** for complete functional traits data.

**Native status.** Although Reference removal plots had higher species richness than Slag removal plots (mean: 8.33 and 4.90, respectively), the latter had a higher proportion of native species (Welch's t-test, $p = 0.0357$), a difference especially apparent between VV-R and VV-S removal plots (**Fig 7A**). Topsoil plots generally had a higher number of nonnative species, likely due to seed contamination, and the difference between Slag and Reference was not significant (Welch's two-sided t-test, $p = 0.3449$; **Fig 7B**). Removal plots on both Reference sites had a higher number of native species than topsoil plots on slag sites ($p < 0.001$). Additional investigation showed that topsoil was a source of nonnative species (see **S1 File**).

## Discussion

Slag sites, among many other fragmented urban habitats with unique environmental conditions, offer insight into the development of novel communities through environmental filtering and adaptation to the urban environment [1]. Community composition and growth of self-assembled vegetation on contaminated sites informs conservation methods such as phytoremediation [3, 6, 7, 20]. Through manipulative experiments and functional trait analysis, we characterized the plant community assemblage on slag from two perspectives: 1) how environmental variables affect plant growth and subsequently succession, and 2) how environmental variables affect plant community assembly. Although the heterogeneity between our locales and blocks within each slag site contributed significantly to cover and biomass results, several consistent patterns were observed. Plant growth and community recovery, measured in percent cover, species number and biomass, were significantly hindered by slag, and community composition on slag did not represent communities on non-slag soil of close proximity. Further studies of community processes on slag would benefit greatly from longer study period, a larger number of soil samples and *in situ* measurements of functional traits.

### Slag effect on plant growth

Plant growth, measured by both percent cover and biomass, was negatively affected by slag. Overall, Slag and Reference sites differed for all measurements of plant growth, both cover and biomass. (**Figs 3** and **5**). Moreover, recovery of Slag removal plots was slower in terms of both

cover and number of recolonized species (**Fig 4**). Combining these results with previous studies that discovered arrested primary succession on natural habitats with similar soil profiles [15, 16], it is reasonable to infer that with such low growth rate, slag vegetation might also experience arrested, or at least delayed primary succession. Linear models showed that environmental variables characterizing slag (category I in **Table 4**) consistently correlated negatively with plant growth. Nevertheless, the small environmental sample size resulted in low $R^2$ and $p$ values (**Table J** in **S1 File**); the patterns could also arise from other sources, including positive association between As and plant growth might be due to a high As content in nutrient-rich commercial topsoil.

In contrast, germination and early growth was not affected by slag. A "time threshold," before which cover and species number did not show a significant site effect, was observed at day 49–63 (**Fig 3**; **Table C-D** in **S1 File**). Previous literature has equivocal evidence on the inhibitory effect of heavy metal ions on seed germination and seedling growth [39–43], though these studies were conducted in a lab setting with one or a few species. Environmental conditions in the field could be more complex, and populations on slag might be more tolerant to heavy metal contamination because of previous exposure.

Additionally, a significant dip in BM-S cover indicates a heat wave on Aug 4 (arrows in **Fig 3A** and **3B**). which also caused a more than 50% mortality of SS on that site (**Fig 5C**). Under such heat, low water retention rate caused by high sand content at BM-S might have resulted in greater desiccation [12].

## Slag effect on community structure

The concept of environmental filtering describes community assembly shaped by environmental conditions of habitats [44, 45]. Theories on environmental filtering predict that environmental factors shape community composition by selecting for species that confer higher fitness, resulting in clustering [46–49]. Studies have shown that this clustering effect is more visible for the functional traits a species possesses than species identity [47], and at larger rather than smaller spatial scales [48–50]. Thus, if slag imposes environmental filtering, species composition or functional traits, or both, should be clustered.

Although species composition of removal plots was more similar within each site, they were different between Slag sites (**Fig 6A** and **Fig B** in **S1 File**). This observation is partly explained by the CCA result (**Fig 6B**), which showed significantly different soil compositions between the two Slag sites. This effect due to locale difference potentially selected for different assemblages in removal plots. The clustering was also observed in functional traits (**Fig 6C**). Graminoid species (grasses, sedges) and species with high SLA aggregated with high K content, which characterizes BM-S in CCA. Graminoids are iconic early-successional species [11, 12], and a high SLA implies high investment in primary production, often a characteristic of fast growing, early successional species [13]. Therefore, BM-S hosted more early-successional species than VV-S. Late-summer flowering species aggregated with high N, Zn and Cr content, all of which characterize VV-S. Late flowering is associated with either competitive or ruderal plant species [51] and has been shown to associate with higher relative reproductive success [52]. Generally, plants colonizing removal plots on Slag were early successional species with high growth rates, consistent with the prediction. Compared to Slag plots, Reference plot species are not characterized by early-successional traits, likely due to regeneration from the existing, later-successional seed bank.

However, it is worth noting that the functional trait data analyzed were obtained from a database, not *in situ*. Therefore, failure to detect strong evidence of environmental filtering might be due to changes of plant functional traits on slag due to plasticity [53, 54]. For

instance, all plants in experimental plots showed slower growth on slag, suggesting traits associated with production such as SLA and LDMC might not be accurately captured by measurements from an online database. Trait measurements through time would provide more information on community assembly, resilience and the importance of functional trait plasticity on slag.

## Implications for conservation

The succession status of slag sites could be important to conservation and management practices. In previous phytoremediation efforts, low survival undermined the ability of plants to take up contaminants [6]. In contrast, studies have shown that industrial ecosystems such as sand-gravel pits, alkaline waste, and limestone quarry floors have been colonized by species suitable for these habitats and show a normal succession process [3, 16, 19, 55]; these authors state that these industrial sites are able to provide habitat for natural successional processes without remediations such as seeding or topsoil capping. Indeed, Smith et al. [19] predicted that a slag dump from 1918 would have accumulated enough organic material to support a pine forest in 75 years, while Řehounková and Prach [20] predicted that 25 years of natural succession would restore gravel-sand pits aged 1–75 years back to grassland, woodland or wetland. However, these conclusions are derived from systems that contain enough organic material [19], where the recovery of community is to the target climax community [20], or when successful establishment of several species can start succession [55]. None of them explicitly addressed the slow growth rate of recolonized species and its potential effect on soil formation argued by Stark et al. [15].

Although many species successfully colonized slag, it remains unknown whether the community would continue along a successional trajectory. Empirical evidence from both germination and the focal species experiments suggests that species on slag developed more slowly than on non-slag soils; slag plots were colonized by fewer species and accumulated less biomass. Therefore, the succession process on slag is expected to be slow. Furthermore, the goal of restoration for slag sites is hard to define. Embedded in the urban matrix, the "natural community" of slag sites is not readily identifiable. In our region of study, the pre-development communities were likely tallgrass prairie, wetlands, or woodlands. Rehabilitation of slag to its "original state" could be difficult, costly, and incur risks due to the introduction of weedy species in commercial topsoil, as we demonstrate here (**Fig C** in **S1 File**).

Nevertheless, urban and industrial habitats might serve as plant refugia, providing an alternative motivation for restoring slag habitats. Because of its similarities to naturally occurring habitats such as alvar or dolomite prairie, slag sites have the potential of hosting rare plants that could only inhabit such environments [3]. For instance, Tomlinson et al. [16] found significant overlap between the flora of alvar habitat and artificial quarry floors through natural colonization. Furthermore, species that are adapted to natural habitats with high metal content, such as serpentine soil or mining sites, could establish in industrial ecosystems [3, 56, 57]. Inhabitable by many competitors, habitats such as slag could provide a refuge for those rare species. A relatively high proportion of native flora and many species with a very high conservation index were identified at both BM-S and VV-S; more strikingly, the latter has a high adjusted FQI, suggesting habitat quality comparable to natural habitats (40.1 compared to 30–40 locally [58–60]; see, however, [61] for a discussion and potential drawbacks of using FQA). Depressions on VV-S hold water during the rainy season, supporting many native fen and marsh species such as *Carex crawei* and *Eleocharis elliptica*; the native orchid *Spiranthes cernua* occurs at both VV-S and BM-S; native plants with high conservation values planted on VV-S, *Bouteloua curtipendula* and *Solidago speciosa*, both displayed 100% survival. These results

suggest that VV-S has potential as an urban refuge for rare plants, especially native species on analogous habitats such as wetlands and dolomite prairie. Thus, habitat reconstructions that mimic a natural analog of slag sites, such as alvar or dolomite prairie [3, 16] might have the greatest success. Those habitats formed by limestone outcrops are native to the Great Lakes region, hosting many species that are adapted to their environments [62–64]. Given the similarity of environmental conditions and many overlapping species such as *Carex spp*., *Eupatorium spp*. and *Panicum virgatum*, *Verbena spp*., *Bouteloua curtipendula*, VV-S could be reclaimed with a plant assemblage that mimics these natural habitats, while retaining its unique species of high conservation values. Future studies may introduce native plants specialized on similar ecosystems to slag and evaluate the feasibility of slag as native plant refugia.

## Conclusions

Although slag has traditionally been viewed as a contaminated wasteland, it is a unique urban-industrial ecosystem that has the potential to host a unique flora. We showed that unfavorable environmental conditions significantly lowered the growth and recovery rates of slag communities in terms of percentage cover, biomass and number of recolonizing species. The composition of slag communities generally corresponds to early successional communities, but the low growth rate may significantly reduce the rate of succession. Unlike many other industrial systems such as sand-gravel pits, natural recovery of slag sites might not be feasible in the short term, requiring active restoration efforts. While topsoil capping might be the most effective method of increasing organic matter, it risks introducing nonnative species, further lowering habitat quality. Although challenging for restoration and remediation, slag could potentially host rare plants from natural, analogous habitats, including flora found at native Midwest habitats such as dolomite prairie and alvar. Therefore, in addition to "radical" remediation such as topsoil capping, burning, and intensive weeding of undesirable species, further efforts to maintain resilient urban ecosystems should also consider retaining slag sites as potential refuges for native plants, which in turn will contribute to the conservation of regional biodiversity.

## Supporting information

**S1 File. Supplemental information: Methods and results.** Detailed history of management for both Reference sites and methods for data analysis, including the list of R packages used. All supplemental results (**Table A**-**J**, **Fig A**-**E** and all associated captions) are also included. A glossary for site, plot and treatment numbering is included.
(DOCX)

**S2 File. Species list from NIRMI plant survey at BM and VV Slag.** "1" denotes presence. C-value stands for Coefficient of Conservatism, based on Universal FQA Calculator.
(CSV)

**S3 File. Species presence/absence for each plot.** Data were compiled from three days of observation: day 77, 91 and 112. Positive identification of a species at any of these days counts as presence, marked by "1". Two unidentified *Dicanthelium* species are marked with *. Species only appeared in topsoil plots are marked with ‡; species germinated from the commercial topsoil in the greenhouse are marked with †.
(CSV)

**S4 File. Functional traits of plant species observed in the survey used for data analysis.** Species abbreviations follow those used in **S3 File**. If one species has multiple measurements of one single trait, measurements were either averaged (numerical) or entered together (categorical). Numerical data with multiple measurements were averaged. Missing data were left blank

and substituted with *NA* in actual analyses. For all analyses, "Type", "Growth Form" and "Native?" were taken out; in addition, "Seedbank Longevity" and "Lateral Spread" were taken out when performing RLQ analysis and fourth-corner method because of low availability of data. Data were unavailable for the two unidentified *Dicanthelium* species and *Cyperus biparti-tus*.
(CSV)

**S5 File. Cover data.** Cover in all removal and control plots is measured from weekly monitoring photos. Time is in days from start of the experiment.
(CSV)

**S6 File. Species number over time.** Species number over the full experimental period in each removal and control plots is obtained from weekly monitoring photos. Time is in days from start of the experiment.
(CSV)

**S7 File. Biomass of germination plots.** Biomass of removal and control plots at the end of experimental period.
(CSV)

**S8 File. Initial biomass of focal species plots.** Biomass of focal species plots at the beginning of experimental period (week 2 after planting).
(CSV)

**S9 File. Final biomass of focal species plots.** Biomass of focal species plots at the end of experimental period.
(CSV)

**S10 File. Reference sites species list.** Species lists were compiled from Chicago Park District seeding plans for BM-R and plant survey on VV-R. Note that these lists were not obtained by the same method used for Slag sites.
(CSV)

## Acknowledgments

We gratefully acknowledge the Chicago Park District and especially Lauren Umek and Stephen Bell for use of the sites and assistance in the field. We thank Mark Bouman and Daniel Spencer for help in selecting potential field sites. We thank Gayle Tonkovich, Samantha King, Samuel Joyce and Erika Yuengling for their help with field work. We thank Sandra Suwanski and John Zdenek for their help with greenhouse work. We thank Stefano Allesina, Dmitry Kondrashov, Marcus Kronforst, Zachary Miller, Brooke Weigel, Timothy Wootton, and members of Pfister-Wootton lab for their comments on data analysis and an earlier version of the manuscript. We gratefully acknowledge Matt A Bahm and two anonymous reviewers for their comments that significantly improved the manuscript.

## Author Contributions

**Conceptualization:** Heng-Xing Zou, Alison E. Anastasio, Catherine A. Pfister.

**Data curation:** Heng-Xing Zou, Alison E. Anastasio.

**Formal analysis:** Heng-Xing Zou, Alison E. Anastasio, Catherine A. Pfister.

**Funding acquisition:** Heng-Xing Zou, Alison E. Anastasio.

**Investigation:** Heng-Xing Zou, Alison E. Anastasio.

**Methodology:** Heng-Xing Zou, Alison E. Anastasio, Catherine A. Pfister.

**Project administration:** Heng-Xing Zou, Alison E. Anastasio.

**Resources:** Alison E. Anastasio.

**Supervision:** Alison E. Anastasio, Catherine A. Pfister.

**Validation:** Heng-Xing Zou, Alison E. Anastasio.

**Visualization:** Heng-Xing Zou.

**Writing – original draft:** Heng-Xing Zou.

**Writing – review & editing:** Heng-Xing Zou, Alison E. Anastasio, Catherine A. Pfister.

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
