## [Decision Letter · Decision Letter 0]

23 Aug 2019

PONE-D-19-19460

Primary Succession on Slag Compared to Urban Soil: A Slower Recovery

PLOS ONE

Dear Dr. Anastasio,

Thank you for submitting your manuscript to PLOS ONE. After careful consideration, we feel that it has merit but does not fully meet PLOS ONE’s publication criteria as it currently stands. Therefore, we invite you to submit a revised version of the manuscript that addresses the points raised during the review process.

I concur with both reviewers comments that the manuscript could be considerably shortened.  You should also address all of the comments provided by the reviewers.  There is valuable information in the manuscript, but the number of analyses conducted seems overdone, especially considering this is a small dataset collected in a single year.  Those limitations should be addressed.  You need to focus your manuscript to one, or only a few areas, and support those with your analysis.  It currently reads as if you tried the opposite, and tried to run many analyses to get them to tell a story.

We would appreciate receiving your revised manuscript by Oct 07 2019 11:59PM. To enhance the reproducibility of your results, we recommend that if applicable you deposit your laboratory protocols in protocols.io, where a protocol can be assigned its own identifier (DOI) such that it can be cited independently in the future. For instructions see: http://journals.plos.org/plosone/s/submission-guidelines#loc-laboratory-protocols

We look forward to receiving your revised manuscript.

Kind regards,

Matt A Bahm, Ph.D.

Academic Editor

PLOS ONE

Journal Requirements:

3. Please ensure that you refer to Figure 1 in your text as, if accepted, production will need this reference to link the reader to the figure.

5. In your Methods section, please provide additional location information of the study areas, including geographic coordinates for the data set if available

6. We note that Figure 1 in your submission contain satellite images which may be copyrighted. All PLOS content is published under the Creative Commons Attribution License (CC BY 4.0), which means that the manuscript, images, and Supporting Information files will be freely available online, and any third party is permitted to access, download, copy, distribute, and use these materials in any way, even commercially, with proper attribution. For these reasons, we cannot publish previously copyrighted maps or satellite images created using proprietary data, such as Google software (Google Maps, Street View, and Earth). For more information, see our copyright guidelines: http://journals.plos.org/plosone/s/licenses-and-copyright.

a) You may seek permission from the original copyright holder of Figure(s) [#] to publish the content specifically under the CC BY 4.0 license. 

Additional Editor Comments (if provided):

As per both reviewers, I agree that the manuscript is too long and should be condensed. You have a small dataset (and only a single growing season) that is likely too limited for some of the multivariate analysis you have conducted, so justification for each analysis should be included. Your use of "Primary succession" is incorrect. These are previously vegetated areas that you removed the vegetation, very clearly secondary succession. If slag had been recently deposited, then you could potentially have had primary succession.

Reviewers' comments:

Reviewer's Responses to Questions

**Comments to the Author**

1. Is the manuscript technically sound, and do the data support the conclusions?

Reviewer #1: Yes

Reviewer #2: Yes

2. Has the statistical analysis been performed appropriately and rigorously? 

Reviewer #1: Yes

Reviewer #2: Yes

3. Have the authors made all data underlying the findings in their manuscript fully available?

Reviewer #1: Yes

Reviewer #2: Yes

4. Is the manuscript presented in an intelligible fashion and written in standard English?

Reviewer #1: Yes

Reviewer #2: Yes

5. Review Comments to the Author

Reviewer #1: Review of

Primary Succession on Slag Compared to Urban Soil: A Slower Recovery

For Applied Soil Ecology (PLOS ONE)

By Zhou, Anastasio, Pfister, University of Chicago

Abstract:

The abstract needs to be more quantitative. What are the differences using diversity indices, richness, or species numbers? What about a few prominent species. Nothing is said about the slag substrate that you used. General statement that it has lots of Ca, Mg, Fe and metals with low pH and water holding capacity. We find slag to have a high pH, not low pH. The general conclusion that slag plots recover more slowly is insufficient without providing some numerical results. Is 1 year a suitable time frame to determine plant succession? The fact that some plants were planted and harvested after a few month's growth is hardly a study of succession.

Introduction:

L42, here you say the pH of slag is generally basic and the pH of the pond with a pH of 12 is evidence of the very high CaO content of slags.

L55-60, very poor citations for the problems you are describing as personal communications Literature is available for the problems you describe so the use of these people as references is not recommended. Plus, you use the words “might” is not useful. And the fact that other sites have been remediated successfully downplays your comments. These sites just haven’t been reclaimed with standard techniques.

L70, “…harsh environment of slag gives diverse plant communities,” continues to erode your hypothesis.

L80, arrested succession is also used in terms of mined land reclamation where current reclamation techniques place compacted soils and heavily-seed aggressive agronomic species for revegetation, these practices do not allow recruitment of native species from the nearby forest or plant communities. See https://arri.osmre.gov/FRA/Advisories/FRA_No.5.pdf

L80, and you should point out that the arrested succession can last for a decade or so and is still relatively short. Primary succession on soiled areas can proceed in a few decades to a closed canopy forest community, so what is the time frame for these slag areas to proceed to a climax vegetation? Has no one studied it on other disturbed areas like sand and gravel quarries, other industrial dumping sites with construction debris like concrete? Oh, I see you make a stab at this in L85-90.

Materials & Methods:

L116, it is extremely important that you tell us when the slag was dumped and how long it has been there. Knowing that one is older than the other is unacceptable.

L132, plant survey and soil test procedures must be in the manuscript, not in some supplemental material. This is the basis of the study!

L149, the soil after vegetation removal was not tilled? How did you remove roots or did you only clip the aboveground biomass?

L152, I really doubt that the removal plots “simulated the recruitment of plants into newly disturbed habitat” since the growing plants would have the advantage with root systems and plants in the seedbank may or may not have had the opportunity to express themselves.

What I conclude is that this study had many measurements and perhaps the authors need to select those that are most important to their story be selected to illustrate and use for results. As of now, the paper is too complex and cumbersome.

Results:

L255, where the corresponding list of species on the reference sites?

L261, do you need to list native and non-native species were 55.6% and 44.4%? How about simply saying native species were 56% at BM-S and 79% at VV-S?

L272, give some soils data! It doesn’t have to be the whole table, but pick some representative elements and give the results without having to go to the supplemental material. Also, you need to summarize it sufficiently so the reader can digest the information clearly to match the text.

L295, why not actually give data in Table 4 and show significance?

L317, no difference in biomass between survived plants between the slag and reference? Explain that. To shorten the paper, this material could be omitted.

L335, “were very low or not significant, due to limited soil data.” This is a poor conclusion and negates the relevance of your study.

L340, perhaps you should reconsider whether to report this information since the reference soil had its own seed bank. Just use the removal plots at the slag and reference sites.

L395, this is some important information, but I wonder if it is real since you said before the soils data were limited in L335.

Discussion

The discussion repeats the results with a few comments.

In my view, it is unnecessary. The few comments can be incorporated into a Results and Discussion section. Omit the discussion.

The “Implications” reads like a discussion, so you can keep that but eliminate the previous part of discussion.

Conclusions

L591, I’m always amused that anyone would think that a slag pile, regardless of age, would in the short term be “restored.” Most industrial or mined or man-made sites are “reclaimed” to a plant community or land use that has environmental and societal value, rather than trying to restore it to some plant community that is presumed to have been there pre-industrialization or that it should suddenly appear due to natural causes. I’m ranting here a bit, but comparing plant community and growth characteristics on these industrial sites without any soil to non-disturbed sites seems to be comparing the growth and characteristics of a pea and an apple; neither is really comparable.

While the study has some value by comparing the species compositions and properties of three distinct “areas” or treatments, are we surprised that a slag dump is recovering in a different way and in the authors terms “slower” than a reference site?

Reviewer #2: This is a well-designed and analyzed study, but I think you need to focus on fewer analyses to get your main point across. The data set is fairly small and while the multivariate analysis methods you explore are quite interesting, the sheer quantity of analysis methods you used distracts from the key differences that you want to emphasize here, as well as the small sample size and design issues that you ran into throughout the season. I think it would help to focus on one analysis for each hypothesis laid out in your introduction, or to split the work into multiple papers within which you could explore the more complex analyses at a deeper level. This is a good exploration into the potential use of slag sites as native species refugia in urban conservation and restoration applications and you also appropriately discuss the need for more data to support some of the inferences suggested by your results here. Overall the writing is clear and consistent; there are some minor typo/grammar and clarity issues that I'm sure will be caught in copy-editing (i.e. in your Abstract you reference slag sites as "low pH").

6. PLOS authors have the option to publish the peer review history of their article (what does this mean?). If published, this will include your full peer review and any attached files.

Reviewer #1: No

Reviewer #2: No

---

## [Author Response · Author response to Decision Letter 0]

1 Oct 2019

Thank you for submitting your manuscript to PLOS ONE. After careful consideration, we feel that it has merit but does not fully meet PLOS ONE’s publication criteria as it currently stands. Therefore, we invite you to submit a revised version of the manuscript that addresses the points raised during the review process.

I concur with both reviewers comments that the manuscript could be considerably shortened. You should also address all of the comments provided by the reviewers. There is valuable information in the manuscript, but the number of analyses conducted seems overdone, especially considering this is a small dataset collected in a single year. Those limitations should be addressed. You need to focus your manuscript to one, or only a few areas, and support those with your analysis. It currently reads as if you tried the opposite, and tried to run many analyses to get them to tell a story.

To enhance the reproducibility of your results, we recommend that if applicable you deposit your laboratory protocols in protocols.io, where a protocol can be assigned its own identifier (DOI) such that it can be cited independently in the future. For instructions see: http://journals.plos.org/plosone/s/submission-guidelines#loc-laboratory-protocols

The majority of our protocols involve standard methods such as biomass and cover measurement. We have specified our field survey protocol in S1 File. We do not think it is necessary to specify other conventional protocols involved in our study.

• A rebuttal letter that responds to each point raised by the academic editor and reviewer(s). This letter should be uploaded as separate file and labeled 'Response to Reviewers'.

• A marked-up copy of your manuscript that highlights changes made to the original version. This file should be uploaded as separate file and labeled 'Revised Manuscript with Track Changes'.

• An unmarked version of your revised paper without tracked changes. This file should be uploaded as separate file and labeled 'Manuscript'.

Journal Requirements:

We have uploaded our supplementary information, including the full data set, to figshare under the following link: https://figshare.com/s/b1f5158a1ea5030d5a92 (private). We will provide public DOI upon formal acceptance.

3. Please ensure that you refer to Figure 1 in your text as, if accepted, production will need this reference to link the reader to the figure.

5. In your Methods section, please provide additional location information of the study areas, including geographic coordinates for the data set if available

6. We note that Figure 1 in your submission contain satellite images which may be copyrighted. All PLOS content is published under the Creative Commons Attribution License (CC BY 4.0), which means that the manuscript, images, and Supporting Information files will be freely available online, and any third party is permitted to access, download, copy, distribute, and use these materials in any way, even commercially, with proper attribution. For these reasons, we cannot publish previously copyrighted maps or satellite images created using proprietary data, such as Google software (Google Maps, Street View, and Earth). For more information, see our copyright guidelines: http://journals.plos.org/plosone/s/licenses-and-copyright.

a) You may seek permission from the original copyright holder of Figure(s) [#] to publish the content specifically under the CC BY 4.0 license. 

>We have reformatted the manuscript according to the above style guidelines. We have also changed Figure 1, which contained information from Google Map, to USGS National Map Viewer [22], and provided coordinates of our sites.

Additional Editor Comments (if provided):

As per both reviewers, I agree that the manuscript is too long and should be condensed. You have a small dataset (and only a single growing season) that is likely too limited for some of the multivariate analysis you have conducted, so justification for each analysis should be included. Your use of "Primary succession" is incorrect. These are previously vegetated areas that you removed the vegetation, very clearly secondary succession. If slag had been recently deposited, then you could potentially have had primary succession.

>We appreciate the efforts by you and the reviewers on the manuscript. In response to your concerns about the length and amount of information in the manuscript, we have substantially shortened it by 55 lines (excluding figure and table captions). Specifically, we now provide a more streamlined results and discussion sections (see below for details). Additionally, we have restated limitations of the study and provided more background information regarding our sites. We realize that our analyses on community structure rely on a relatively small data set. However, we argue that these environmental filtering and trait data should be kept in the narrative because they provide essential information on slag communities in addition to species lists and Floristic Quality Assessments. We have improved the presentation of community structure results by compiling a new Fig 6 from NMDS results of removal plots only (originally Fig 6A), CCA and fourth-corner results (originally Fig 7B, 7C). We have included a new Fig 7, originally in Supplemental Information, in the main text to support our description of native status of species recolonized in experimental plots.

We acknowledge that our experimental simulations do not fully reflect primary succession in nature, but they did provide good approximations of early successional habitats with little topsoil and no current residents. We believe that this treatment is able to provide information that could be used to analyze the actual primary succession process, namely recovery rate. According to our results, both the growth rate (cover and biomass) and recolonization rate (species number) in removal plots on slag were lower than plots on non-slag soil. Because the two processes are essential to community assembly on newly disturbed habitat, it is reasonable to conclude that primary succession would be slower on slag than on non-slag soil. We have clarified this point in title and abstract by substituting “primary succession” by recovery, which was directly measured in our experiments.

To better frame the main thesis of our paper, we changed the title to “Early Succession on Slag Compared to Urban Soil: A Slower Recovery.”

Reviewers' comments:

Reviewer's Responses to Questions

Comments to the Author

1. Is the manuscript technically sound, and do the data support the conclusions?

Reviewer #1: Yes

Reviewer #2: Yes

2. Has the statistical analysis been performed appropriately and rigorously? 

Reviewer #1: Yes

Reviewer #2: Yes

3. Have the authors made all data underlying the findings in their manuscript fully available?

Reviewer #1: Yes

Reviewer #2: Yes

4. Is the manuscript presented in an intelligible fashion and written in standard English?

Reviewer #1: Yes

Reviewer #2: Yes

5. Review Comments to the Author

Reviewer #1: Review of

Primary Succession on Slag Compared to Urban Soil: A Slower Recovery

For Applied Soil Ecology (PLOS ONE)

By Zhou, Anastasio, Pfister, University of Chicago

Abstract:

The abstract needs to be more quantitative. What are the differences using diversity indices, richness, or species numbers? What about a few prominent species. Nothing is said about the slag substrate that you used. General statement that it has lots of Ca, Mg, Fe and metals with low pH and water holding capacity. We find slag to have a high pH, not low pH. The general conclusion that slag plots recover more slowly is insufficient without providing some numerical results. Is 1 year a suitable time frame to determine plant succession? The fact that some plants were planted and harvested after a few month's growth is hardly a study of succession.

>We appreciate that the reviewer asked for more accurate wording regarding succession. We have substituted “primary succession” to “recovery”, which was directly measured in our experiments. We agree that with such short experimental period it would be impossible to assess a complete successional trajectory; the experiment was conducted to characterize recovery rate, which is important for primary succession process. 

We now include more quantitative information in the abstract. Recovery rate was characterized by 3 variables (cover, biomass and species numbers), two of which were time series data. From these, we estimated a maximal difference in cover between Slag and Reference removal plots, percentage of native species on one of our slag sites and also described the soil characteristics of our sites more specifically. 

Introduction:

L42, here you say the pH of slag is generally basic and the pH of the pond with a pH of 12 is evidence of the very high CaO content of slags.

>Added. Given the high pH and Ca content, it is very likely that slag contains remnant CaO from crude processing of iron ore. 

L55-60, very poor citations for the problems you are describing as personal communications Literature is available for the problems you describe so the use of these people as references is not recommended. Plus, you use the words “might” is not useful. And the fact that other sites have been remediated successfully downplays your comments. These sites just haven’t been reclaimed with standard techniques.

>Thank you for the comment. We added two more citations to further back up the point that common methods for remediation are not always successful (L55-57). We have cleared the ambiguity that some sites have been successfully restored; they are only reclaimed to different forms such as urban parks and recreational areas, rather than forms with higher ecological values such as natural areas or wildlife refuges (L53-55). 

L70, “…harsh environment of slag gives diverse plant communities,” continues to erode your hypothesis.

>We agree that this statement might appear contrary to our results. However, slag did have characteristics of harsh soil environments (Table 3), and our results indicate that substantial diversity still colonizes these slag soils.

L80, arrested succession is also used in terms of mined land reclamation where current reclamation techniques place compacted soils and heavily-seed aggressive agronomic species for revegetation, these practices do not allow recruitment of native species from the nearby forest or plant communities. See https://arri.osmre.gov/FRA/Advisories/FRA_No.5.pdf

>Thank you for pointing out this useful source. We have added two additional sources (including this one) to better illustrate the idea of “arrested succession.” (L84-86)

L80, and you should point out that the arrested succession can last for a decade or so and is still relatively short. Primary succession on soiled areas can proceed in a few decades to a closed canopy forest community, so what is the time frame for these slag areas to proceed to a climax vegetation? Has no one studied it on other disturbed areas like sand and gravel quarries, other industrial dumping sites with construction debris like concrete? Oh, I see you make a stab at this in L85-90.

>We have clarified this point by addressing the general timeframe set by previous studies [19, 20]. Both studies predict that the slag community would achieve climax after 75-150 years (L90-91).

Materials & Methods:

L116, it is extremely important that you tell us when the slag was dumped and how long it has been there. Knowing that one is older than the other is unacceptable.

>We have added relevant information from a 1997 USGS report [5]. We have also obtained information on Reference sites from Chicago Park District, specifically how they were managed (see S1 File).

L132, plant survey and soil test procedures must be in the manuscript, not in some supplemental material. This is the basis of the study!

>We have added plant survey and soil test procedures back to the manuscript (L142-161).

L149, the soil after vegetation removal was not tilled? How did you remove roots or did you only clip the aboveground biomass?

>We have further clarified our plot preparation methods. Aboveground biomass and roots were removed, and the top 5-10 mm layer of soil was tilled manually (L179-182). 

L152, I really doubt that the removal plots “simulated the recruitment of plants into newly disturbed habitat” since the growing plants would have the advantage with root systems and plants in the seedbank may or may not have had the opportunity to express themselves.

>We agree that previous residents could have unpredictable effects on species recolonizing removal plots. However, because all plant material was removed as completely as possible, it be considered a newly disturbed and relatively “open” habitat for recolonization. Even if pre-existing plants were advantageous in recolonizing removal plots, the overall recovery rate still showed a significant difference between Slag and Reference sites. The potential advantage did not affect the conclusion that Slag plots recovered more slowly.

What I conclude is that this study had many measurements and perhaps the authors need to select those that are most important to their story be selected to illustrate and use for results. As of now, the paper is too complex and cumbersome.

>We share your concerns that we need to select parts that are most relevant to our narrative. Please see below for specific edits.

Results:

L255, where the corresponding list of species on the reference sites?

>Unfortunately, complete plant survey data were not available at Reference sites. We have, however, obtained a list of species seeded to BM-R, one of our Reference sites, and data from a quick survey on VV-R, the other Reference site, both by Chicago Park District. Although we have compiled them into species lists and attached as Supplemental Information (S10 File), we were unable to repeat our analyses because those lists were obtained through different methodologies and therefore are in different resolution; furthermore, applying FQA to a seeding mix, which aims for habitat restoration, is less meaningful. We have explained this at the end of Results: Plant Survey of Slag Sites (L291-294).

L261, do you need to list native and non-native species were 55.6% and 44.4%? How about simply saying native species were 56% at BM-S and 79% at VV-S?

>Great point. We have changed our presentation of that result accordingly (L286-288).

L272, give some soils data! It doesn’t have to be the whole table, but pick some representative elements and give the results without having to go to the supplemental material. Also, you need to summarize it sufficiently so the reader can digest the information clearly to match the text.

>Thank you for the suggestion. We have put a soil result table in the manuscript; the full version is still in supplemental information. We have also added a short word description of results (L297-303).

L295, why not actually give data in Table 4 and show significance?

>We appreciate your suggestion. However, Table 4 (now Table 5 in revised manuscript) shows summarized statistics from ANOVA of site and locale effects of percent cover. The raw data (S5 File) is a time series of cover from all 32 germination plots. Due to the sheer amount of data we couldn’t present it all in the main text. 

L317, no difference in biomass between survived plants between the slag and reference? Explain that. To shorten the paper, this material could be omitted.

>The original statement, “if all zeroes caused by mortality were removed, no significant difference in SS biomass was detected between BM-S and VV-S,” implies that the difference between biomass of SS on Slag and Reference using a data set including all zeroes was mainly due to the high mortality (the presence of zeroes.) However, survived SS plants that survived weighted less on Slag than on Reference, suggesting slag negatively impacted SS growth. Because this statement was confusing and redundant, we have removed it in revised manuscript.

L335, “were very low or not significant, due to limited soil data.” This is a poor conclusion and negates the relevance of your study.

>Agreed. We removed this statement.

L340, perhaps you should reconsider whether to report this information since the reference soil had its own seed bank. Just use the removal plots at the slag and reference sites.

>We appreciate your suggestion. The extra information we provided was necessary because most community structure analyses were conducted using removal data only. We have moved the few analyses using topsoil plot data, as well as detailed information regarding existing seed bank in topsoil, to Supplementary Information (S1 File).

L395, this is some important information, but I wonder if it is real since you said before the soils data were limited in L335.

>Thank you for pointing it out. The result is statistically significant, but it does not necessarily apply to contexts out of this study. We have expanded this point in Discussion.

Discussion

The discussion repeats the results with a few comments.

In my view, it is unnecessary. The few comments can be incorporated into a Results and Discussion section. Omit the discussion.

The “Implications” reads like a discussion, so you can keep that but eliminate the previous part of discussion.

>We share with your concern that Results and Discussion have many overlapping contents. Nevertheless, a separate Discussion section allowed us to more fully experimental and analytical results. However, we have streamlined Results by only including descriptions, leaving all interpretation in Discussion. We have also renamed and reordered subtitles in Results such that they are more descriptive and correspond to the three hypotheses at the end of Introduction. Additionally, we have reduced reiterations of results in Discussion and focused more on implications of results and potential drawbacks of our methods. For instance, we have stressed our limitations due to low sample size and in situ functional data availability, shortened the introduction on environmental filtering by providing only information relevant to our results, and cleaned up the discussion on environmental filtering results to better reflect our hypotheses by removing tangential contents on metacommunities. For a more comprehensive perspective on FQA, we have cited a recent perspective paper with detailed discussion of this metric. We have also improved our use of language to present concise information.

Conclusions

L591, I’m always amused that anyone would think that a slag pile, regardless of age, would in the short term be “restored.” Most industrial or mined or man-made sites are “reclaimed” to a plant community or land use that has environmental and societal value, rather than trying to restore it to some plant community that is presumed to have been there pre-industrialization or that it should suddenly appear due to natural causes. I’m ranting here a bit, but comparing plant community and growth characteristics on these industrial sites without any soil to non-disturbed sites seems to be comparing the growth and characteristics of a pea and an apple; neither is really comparable.

While the study has some value by comparing the species compositions and properties of three distinct “areas” or treatments, are we surprised that a slag dump is recovering in a different way and in the authors terms “slower” than a reference site?

>We appreciate the motivation for these comments. We have clarified, in the Introduction, that our sites are plain, post-industrial landscapes covered by a layer of slag and have gone through substantial time yet still exhibit early successional patterns (L50-68). We would also like to restate that our Reference sites also have complex land use history and are very different from remnant natural area. We realize that “restoration” could be used in a very specific sense that implies “restoring” a habitat to its “original state”. We have clarified that our aim of this study was not to evaluate whether “restoration” is feasible for slag sites, but to suggest an alternative reclamation method based on our understanding of the specific study system. Thus, we expected a slower recovery rate of slag communities after disturbance and suggested an alternative approach of transforming slag sites to native plant refugia based on this slow recovery rate and our plant survey results (L273-294). 

Reviewer #2: This is a well-designed and analyzed study, but I think you need to focus on fewer analyses to get your main point across. The data set is fairly small and while the multivariate analysis methods you explore are quite interesting, the sheer quantity of analysis methods you used distracts from the key differences that you want to emphasize here, as well as the small sample size and design issues that you ran into throughout the season. I think it would help to focus on one analysis for each hypothesis laid out in your introduction, or to split the work into multiple papers within which you could explore the more complex analyses at a deeper level. This is a good exploration into the potential use of slag sites as native species refugia in urban conservation and restoration applications and you also appropriately discuss the need for more data to support some of the inferences suggested by your results here. Overall the writing is clear and consistent; there are some minor typo/grammar and clarity issues that I'm sure will be caught in copy-editing (i.e. in your Abstract you reference slag sites as "low pH").

>We appreciate the positive comments from Reviewer #2 and believe we have sufficiently addressed their major points – that the manuscript is too long and there are too many analyses – in our responses above. We have improved our presentation of results by reordering and renaming subtitles of Results section: plant surveys and soil test results for the first hypothesis, slag effects on recovery (cover, biomass, species number, linear regression) for the second and structure of recolonized species (CCA and fourth-corner) for the third. We do feel the need of presenting our results in one coherent paper because it provides a more complete narrative on the topic.

6. PLOS authors have the option to publish the peer review history of their article (what does this mean?). If published, this will include your full peer review and any attached files.

Do you want your identity to be public for this peer review? For information about this choice, including consent withdrawal, please see our Privacy Policy.

Reviewer #1: No

Reviewer #2: No

---

## [Decision Letter · Decision Letter 1]

9 Oct 2019

Early Succession on Slag Compared to Urban Soil: A Slower Recovery

PONE-D-19-19460R1

Dear Dr. Anastasio,

We are pleased to inform you that your manuscript has been judged scientifically suitable for publication and will be formally accepted for publication once it complies with all outstanding technical requirements.

With kind regards,

Matt A Bahm, Ph.D.

Academic Editor

PLOS ONE

Additional Editor Comments (optional):

Reviewers' comments:

Reviewer's Responses to Questions

**Comments to the Author**

1. If the authors have adequately addressed your comments raised in a previous round of review and you feel that this manuscript is now acceptable for publication, you may indicate that here to bypass the “Comments to the Author” section, enter your conflict of interest statement in the “Confidential to Editor” section, and submit your "Accept" recommendation.

Reviewer #1: All comments have been addressed

2. Is the manuscript technically sound, and do the data support the conclusions?

Reviewer #1: Yes

3. Has the statistical analysis been performed appropriately and rigorously? 

Reviewer #1: Yes

4. Have the authors made all data underlying the findings in their manuscript fully available?

Reviewer #1: Yes

5. Is the manuscript presented in an intelligible fashion and written in standard English?

Reviewer #1: Yes

6. Review Comments to the Author

Reviewer #1: It appears that the authors have addressed my comments and concerns.

7. PLOS authors have the option to publish the peer review history of their article (what does this mean?). If published, this will include your full peer review and any attached files.

Reviewer #1: No

---

## [Editor Report · Acceptance letter]

23 Oct 2019

PONE-D-19-19460R1 

Early Succession on Slag Compared to Urban Soil: A Slower Recovery 

Dear Dr. Anastasio:

I am pleased to inform you that your manuscript has been deemed suitable for publication in PLOS ONE. Congratulations! Your manuscript is now with our production department. 

With kind regards,

on behalf of

Dr. Matt A Bahm 

Academic Editor

PLOS ONE